# Learning from Unlabelled Videos Using Contrastive Predictive Neural 3D Mapping

**Adam W. Harley**
Carnegie Mellon University
aharley@cmu.edu

**Shrinidhi K. Lakshmikanth**
Carnegie Mellon University
kowshika@cmu.edu

**Fangyu Li**
Carnegie Mellon University
fangyul@cmu.edu

**Xian Zhou**
Carnegie Mellon University
zhouxian@cmu.edu

**Hsiao-Yu Fish Tung**
Carnegie Mellon University
htung@cs.cmu.edu

**Katerina Fragkiadaki**
Carnegie Mellon University
katef@cs.cmu.edu

## Abstract

Predictive coding theories suggest that the brain learns by predicting observations at various levels of abstraction. One of the most basic prediction tasks is view prediction: how would a given scene look from an alternative viewpoint? Humans excel at this task. Our ability to imagine and fill in missing information is tightly coupled with perception: we *feel* as if we see the world in 3 dimensions, while in fact, information from only the front surface of the world hits our retinas. This paper explores the role of view prediction in the development of 3D visual recognition. We propose neural 3D mapping networks, which take as input 2.5D (color and depth) video streams captured by a moving camera, and lift them to stable 3D feature maps of the scene, by disentangling the scene content from the motion of the camera. The model also projects its 3D feature maps to novel viewpoints, to predict and match against target views. We propose contrastive prediction losses to replace the standard color regression loss, and show that this leads to better performance on complex photorealistic data. We show that the proposed model learns visual representations useful for (1) semi-supervised learning of 3D object detectors, and (2) unsupervised learning of 3D moving object detectors, by estimating the motion of the inferred 3D feature maps in videos of dynamic scenes. To the best of our knowledge, this is the first work that empirically shows view prediction to be a scalable self-supervised task beneficial to 3D object detection.

## 1 Introduction

Predictive coding theories (Rao & Ballard, 1999; Friston, 2003) suggest that the brain learns by predicting observations at various levels of abstraction. These theories currently have extensive empirical support: stimuli are processed more quickly if they are predictable (McClelland & Rumelhart, 1981; Pinto et al., 2015), prediction error is reflected in increased neural activity (Rao & Ballard, 1999; Brodski et al., 2015), and disproven expectations lead to learning (Schultz et al., 1997). A basic prediction task is view prediction: from one viewpoint, predict what the scene would look like from another viewpoint. Learning this task does not require supervision from any annotations; supervision is freely available to a mobile agent in a 3D world who can estimate its egomotion (Patla, 1991). Humans excel at this task: we can effortlessly imagine plausible hypotheses for the occluded side of objects in a photograph, or guess what we would see if we walked around our office desks. Our ability to *imagine*

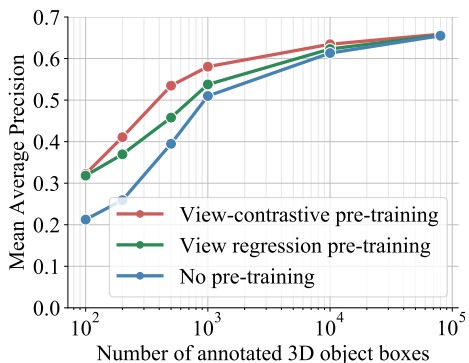

Figure 1: **Semi-supervised 3D object detection.** Pre-training with view-contrastive prediction improves results, especially when there are few 3D bounding box annotations.

information missing from the current image view—and necessary for predicting alternative views—is tightly coupled with visual perception. We infer a mental representation of the world that is 3-dimensional, in which the objects are distinct, have 3D extent, occlude one another, and so on. Despite our 2-dimensional visual input, and despite never having been supplied a bounding box or segmentation mask as supervision, our ability for 3D perception emerges early in infancy (Spelke et al., 1982; Soska & Johnson, 2008).

In this paper, we explore the link between view predictive learning and the emergence of 3D perception in computational models of perception, on mobile agents in static and dynamic scenes. Our models are trained to predict views of static scenes given 2.5D (color and depth; RGB-D) video streams as input, and are evaluated on their ability to detect objects in 3D. Our models map the 2.5D input streams into 3D feature volumes of the depicted scene. At every frame, the architecture estimates and accounts for the motion of the camera, so that the internal 3D representation remains stable under egomotion. The model projects its inferred 3D feature maps to novel viewpoints, and matches them against visual representations extracted from the target view. We propose contrastive losses to measure the match error, and backpropagate gradients end-to-end in our differentiable modular architecture. At test time, our model forms plausible 3D completions of the scene given RGB-D video streams or even a *single RGB-D image* as input: it learns to fill in information behind occlusions, and infer the 3D extents of objects.

We evaluate the trained 3D representations in two tasks. **(1)** Semi-supervised learning of 3D object detectors (Figure 1): We show that view-contrastive pre-training helps detect objects in 3D, especially in the low-annotations regime. **(2)** Unsupervised 3D moving object detection (Figure 3-right): Our model can detect moving objects in 3D without any human annotations, by forming a 3D feature volume for each timestep, then estimating the motion field between volumes, and clustering the motion into objects.

We have the following contributions over prior works:

1. **Novel view-contrastive prediction objectives**: We show that our contrastive losses outperform RGB regression (Dosovitskiy et al., 2017; Tung et al., 2019) and variational auto-encoder (VAE) alternatives (Eslami et al., 2018) in semi-supervised 3D object detection.

2. **A novel unsupervised 3D moving object detection method**: By estimating 3D motion inside egomotion-stabilized 3D feature maps, we can detect moving objects unsupervised, outperforming 2.5D baselines and iterative generative what-where VAEs of previous works (Kosiorek et al., 2018; Hsieh et al., 2018).

3. **Simulation-to-real transfer of the acquired 3D feature representations**: We show that view-contrastive pre-training in simulation boosts the performance of 3D object detection in real data.

Our code and data are publicly available[1].

## 2 RELATED WORK

**View prediction** View prediction has been the center of much recent research effort. Most methods test their models in single-object scenes, and aim to generate beautiful images for graphics applications (Kar et al., 2017; Sitzmann et al., 2019; Tatarchenko et al., 2016; Saito et al., 2019), as opposed to learning general-purpose visual representations. In this work, we use view prediction to help object detection, not the inverse. The work of Eslami et al. (2018) attempted view prediction in full scenes, yet only experimented with toy data containing a few colored 3D blocks. Their model cannot effectively generalize beyond the training distribution, e.g., cannot generalize across scenes of variable numbers of objects. The work of Tung et al. (2019) is the closest to our work. Their model is also an inverse graphics network equipped with a 3-dimensional feature bottleneck, trained for view prediction; it showed strong generalization across scenes, number of objects, and arrangements. However, the authors demonstrated its abilities only in toy simulated scenes, similar to those used in Eslami et al. (2018). Furthermore, they did not evaluate the usefulness of the learned features for a downstream semantic task (beyond view prediction). This raises questions on the scalability and usefulness of view prediction as an objective for self-supervised visual representation learning, which our work aims to address. We compare against the state-of-the-art model of Tung et al. (2019)

---

[1] https://github.com/aharley/neural_3d_mapping

and show that the features learned under our proposed view-contrastive losses are more semantically meaningful (Figure 1). To the best of our knowledge, this is the first work that can discover objects in 3D from a single camera viewpoint, without any human annotations of object boxes or masks.

**Predictive visual feature learning**  Predictive coding theories suggest that much of the learning in the brain is of a predictive nature (Rao & Ballard, 1999; Friston, 2003). Recent work in unsupervised learning of word representations has successfully used ideas of predictive coding to learn word representations by predicting neighboring words (Mikolov et al., 2013). Many challenges emerge in going from a finite-word vocabulary to the continuous high-dimensional image data manifold. Unimodal losses such as mean squared error are not very useful when predicting high dimensional data, due to the stochasticity of the output space. Researchers have tried to handle such stochasticity using latent variable models (Loehlin, 1987) or autoregressive prediction of the output pixel space, which involves sampling each pixel value from a categorical distribution conditioned on the output thus far (Van den Oord et al., 2016). Another option is to make predictions in a feature space which is less stochastic than the input. Recently, Oord et al. (2018) followed this direction and used an objective that preserves the mutual information between "top-down" contextual features predicted from input observations, and "bottom-up" features produced from future observations; it applied this objective in speech, text, and image crops. The view-contrastive loss proposed in this work is a non-probabilistic version of their contrastive objective. However, our work focuses on the video domain as opposed to image patches, and uses drastically different architectures for both the top-down and bottom-up representations, involving a 3D egomotion-stabilized bottleneck.

## 3  CONTRASTIVE PREDICTIVE NEURAL 3D MAPPING

We consider a mobile agent that can move about the scene at will. The agent has a color camera with known intrinsics, and a depth sensor registered to the camera's coordinate frame. At training time, the agent has access to its camera pose, and it learns in this stage to imagine full 3D scenes (via view prediction), and to estimate egomotion (from ground-truth poses). It is reasonable to assume that a mobile agent who moves at will has access to its approximate egomotion, since it chooses where to move and what to look at (Patla, 1991). Active vision is outside the scope of this work, so our agent simply chooses viewpoints randomly. At test time, the model estimates egomotion on-the-fly from its RGB-D inputs. We use groundtruth depth provided by the simulation environment, and we will show in Sec. 4 that the learned models generalize to the real world, where (sparser) depth is provided by a LiDAR unit. We describe our model architecture in Sec. 3.1, our view-contrastive prediction objectives in Sec. 3.2, and our unsupervised 3D object segmentation method in Sec. 3.3.

### 3.1  NEURAL 3D MAPPING

Our model's architecture is illustrated in Figure 2-left. It is a recurrent neural network (RNN) with a memory state tensor $\mathbf{M}^{(t)} \in \mathbb{R}^{w \times h \times d \times c}$, which has three spatial dimensions (width $w$, height $h$, and depth $d$) and a feature dimension ($c$ channels per grid location). The latent state aims to capture an informative and geometrically-consistent 3D deep feature map of the world space. Therefore, the spatial extents correspond to a large cuboid of world space, defined with respect to the camera's position at the first timestep. We refer to the latent state as the model's *imagination* to emphasize that most of the grid locations in $\mathbf{M}^{(t)}$ will not be observed by any sensor, and so the feature content must be "imagined" by the model.

Our model is made up of differentiable modules that go back and forth between 3D feature imagination space and 2D image space. It builds on the recently proposed geometry-aware recurrent neural networks (GRNNs) of Tung et al. (2019), which also have a 3D egomotion-stabilized latent space, and are trained for RGB prediction. Our model can be considered a type of GRNN. In comparison to Tung et al. (2019): **(1)** our egomotion module can handle general camera motion, as opposed to a 2-degree-of-freedom sphere-locked camera. This is a critical requirement for handling data that comes from freely-moving cameras, such as those mounted on mobile vehicles, as opposed to only orbiting cameras. **(2)** Our 3D-to-2D projection module decodes the 3D map into 2D feature maps, as opposed to RGB images, and uses view-contrastive prediction as the objective, as opposed to regression. In Table 3 and Figure 8, we show that nearest neighbors in our learned feature space are more semantically related than neighbors delivered by RGB regression. We briefly describe each neural module of our architecture next. Implementation details are in the appendix.

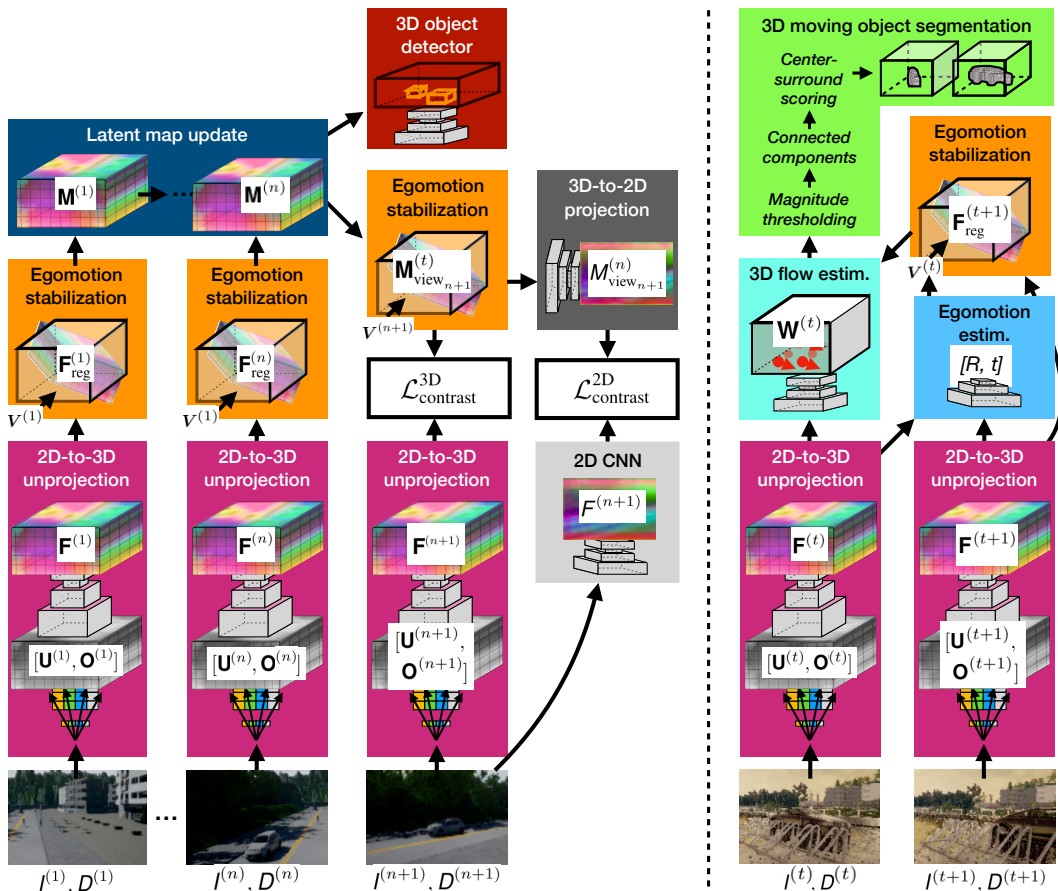

Figure 2: **Contrastive predictive neural 3D mapping.** *Left:* Learning visual feature representations by moving in static scenes. The neural 3D mapper learns to lift 2.5D video streams to egomotion-stabilized 3D feature maps of the scene by optimizing for view-contrastive prediction. *Right:* Learning to segment 3D moving objects by watching them move. Non-zero 3D motion in the egomotion-stabilized 3D feature space reveals independently moving objects and their 3D extent, without any human annotations.

**2D-to-3D unprojection** This module converts the input RGB image $I^{(t)} \in \mathbb{R}^{w \times h \times 3}$ and pointcloud $D^{(t)} \in \mathbb{R}^{n \times 3}$ into 3D tensors. The RGB is "unprojected" into a 3D tensor $\mathbf{U}^{(t)} \in \mathbb{R}^{w \times h \times d \times 3}$ by filling each 3D grid location with the RGB value of its corresponding subpixel. The pointcloud is converted to a 3D occupancy grid $\mathbf{O}^{(t)} \in \mathbb{R}^{w \times h \times d \times 1}$, by assigning each voxel a value of 1 or 0, depending on whether or not a point lands in the voxel. We then convert the concatenation of these tensors into a 3D feature tensor $\mathbf{F}^{(t)} \in \mathbb{R}^{w \times h \times d \times c}$, via a 3D convolutional encoder-decoder network with skip connections. We $L_2$-normalize the feature in each grid cell.

**Egomotion estimation** This module computes the relative 3D rotation and translation between the current camera pose (at time $t$) and the reference pose (from time 1), allowing us to warp the feature tensor $\mathbf{F}^{(t)}$ into a registered version $\mathbf{F}^{(t)}_{\mathrm{reg}}$. In principle any egomotion estimator could be used here, but we find that our 3D feature tensors are well-suited to a 3D coarse-to-fine alignment search, similar to the 2D process in the state-of-the-art optical flow model PWC-Net (Sun et al., 2018). Given the 3D tensors of the two timesteps, $\mathbf{F}^{(1)}$ and $\mathbf{F}^{(t)}$, we incrementally warp $\mathbf{F}^{(t)}$ into alignment with $\mathbf{F}^{(1)}$, by estimating the approximate transformation at a coarse scale, then estimating residual transformations at finer scales. This is done efficiently with 6D cross correlations and cost volumes. Following PWC-Net, we use fully-connected layers to convert the cost volumes into motion estimates. While our neural architecture is trained end-to-end to optimize a view prediction objective, our egomotion module by exception is trained supervised using pairs of frames with annotated egomotion. In this way, it learns to be invariant to moving objects in the scene.

**Latent map update** This module aggregates egomotion-stabilized (registered) feature tensors into the memory tensor $\mathbf{M}^{(t)}$. On the first timestep, we set $\mathbf{M}^{(1)} = \mathbf{F}^{(1)}$. On later timesteps, we update the memory with a simple running average.

**3D-to-2D projection** This module "renders" the 3D feature state $\mathbf{M}^{(t)}$ into a 2D feature map of a desired viewpoint $\boldsymbol{V}^{(k)}$. We first warp the 3D feature map $\mathbf{M}^{(t)}$ into a view-aligned version $\mathbf{M}^{(t)}_{\text{view}_k}$, then map it to a 2D feature map $M^{(t)}_{\text{view}_k}$ with a 2-block 2D ResNet (He et al., 2016).

**3D object detection** Given images with annotated 3D object boxes, we train a 3D object detector that takes as input the 3D feature map $\mathbf{M}^{(t)}$, and outputs 3D bounding boxes for the objects present. Our object detector is a 3D adaptation of the state-of-the-art 2D object detector, Faster-RCNN (Ren et al., 2015). The model outputs 3D axis-aligned boxes with objectness confidences.

## 3.2 CONTRASTIVE PREDICTIVE TRAINING

Given a set of input RGB images $(I^{(1)}, \ldots, I^{(n)})$, pointclouds $(D^{(1)}, \ldots, D^{(n)})$, and camera poses $(\boldsymbol{V}^{(1)}, \ldots, \boldsymbol{V}^{(n)})$, we train our model to predict feature abstractions of an unseen input $(I^{(n+1)}, D^{(n+1)}, \boldsymbol{V}^{(n+1)})$, as shown in Figure 2-left. We consider two types of representations for the target view: a top-down one, $\mathcal{T} = f[(I^{(1)}, D^{(1)}, \boldsymbol{V}^{(1)}), \ldots, (I^{(n)}, D^{(n)}, \boldsymbol{V}^{(n)}), \boldsymbol{V}^{(n+1)}]$, where $f$ is the top-down feature prediction process, and a bottom-up one, $\mathcal{B} = g[I^{(n+1)}, D^{(n+1)}]$, where $g$ is the bottom-up feature extraction process. Note that the top-down representation has access to the viewpoint $\boldsymbol{V}^{(n+1)}$ but not to observations from that viewpoint $(I^{(n+1)}, D^{(n+1)})$, while the bottom-up representation is only a function of those observations.

We construct 2D and 3D versions of these representation types, using our architecture modules:

- We obtain $\mathcal{T}^{\text{3D}} = \mathbf{M}^{(n)}$ by encoding the set of inputs from indices $1, \ldots, n$.
- We obtain $\mathcal{B}^{\text{3D}} = \mathbf{F}^{(n+1)}$ by encoding the single input from index $n + 1$.
- We obtain $\mathcal{T}^{\text{2D}} = M^{(n)}_{\text{view}_{n+1}}$ by rendering $\mathbf{M}^{(n)}$ from viewpoint $\boldsymbol{V}^{(n+1)}$.
- We obtain $\mathcal{B}^{\text{2D}} = F^{(n+1)}$ by convolving $I^{(n+1)}$ with a 3-block 2D ResNet (He et al., 2016).

Finally, the contrastive losses pull corresponding (top-down and bottom-up) features close together in embedding space, and push non-corresponding ones beyond a margin of distance:

$$\mathcal{L}^{\text{2D}}_{\text{contrast}} = \sum_{i,j,m,n} \max\left(\mathcal{Y}^{\text{2D}}_{ij,mn}(\|\mathcal{T}^{\text{2D}}_{ij} - \mathcal{B}^{\text{2D}}_{mn}\|_2 - \alpha), 0\right), \tag{1}$$

$$\mathcal{L}^{\text{3D}}_{\text{contrast}} = \sum_{i,j,k,m,n,o} \max\left(\mathcal{Y}^{\text{3D}}_{ijk,mno}(\|\mathcal{T}^{\text{3D}}_{ijk} - \mathcal{B}^{\text{3D}}_{mno}\|_2 - \alpha), 0\right), \tag{2}$$

where $\alpha$ is the margin size, and $\mathcal{Y}$ is 1 at indices where $\mathcal{T}$ corresponds to $\mathcal{B}$, and $-1$ everywhere else. The losses ask tensors depicting the same scene, but acquired from different viewpoints, to contain the same features. The performance of a metric learning loss depends heavily on the sampling strategy used (Schroff et al., 2015; Song et al., 2016; Sohn, 2016). We use the distance-weighted sampling strategy proposed by Wu et al. (2017) which uniformly samples "easy" and "hard" negatives; we find this outperforms both random sampling and semi-hard sampling (Schroff et al., 2015).

## 3.3 UNSUPERVISED 3D MOVING OBJECT DETECTION

Upon training, our model learns to map even a *single* RGB-D input to a complete 3D imagination, as we show in Figure 2-right. Given two temporally consecutive and egomotion-stabilized 3D maps $\mathbf{F}^{(t)}, \mathbf{F}^{(t+1)}_{\text{reg}}$, predicted independently using inputs $(I^{(t)}, D^{(t)})$ and $(I^{(t+1)}, D^{(t+1)})$, we train a motion estimation module to predict the 3D motion field $\mathbf{W}^{(t)}$ between them, which we call 3D imagination flow. Since we have accounted for camera motion, this 3D motion field should only be non-zero for independently moving objects. We obtain 3D object proposals by clustering the 3D flow vectors, extending classic motion clustering methods (Brox & Malik, 2010; Ochs & Brox, 2011) to an egomotion-stabilized 3D feature space, as opposed to 2D pixel space.

### 3.3.1 Estimating 3D Imagination Flow

Our 3D "imagination flow" module is a 3D adaptation of the PWC-Net 2D optical flow model (Sun et al., 2018). Note that our model only needs to estimate motion of the independently-moving part of the scene, since egomotion has been accounted for. It works by iterating across scales in a coarse-to-fine manner; at each scale, we compute a 3D cost volume, convert these costs to 3D displacement vectors, and incrementally warp the two tensors to align them. We use two self-supervised tasks:

1. Synthetic rigid transformations: We apply random rotations and translations to $\mathbf{F}^{(t)}$ and ask the model to recover the dense 3D flow field that corresponds to the transformation.
2. Unsupervised 3D temporal feature matching:

$$\mathcal{L}_{\text{warp}} = \sum_{i,j,k} ||\mathbf{F}^{(t)}_{i,j,k} - \mathcal{W}(\mathbf{F}^{(t+1)}_{\text{reg}}, \mathbf{W}^{(t)})_{i,j,k}||, \tag{3}$$

where $\mathcal{W}(\mathbf{F}^{(t+1)}, \mathbf{W}^{(t)})$ back-warps $\mathbf{F}^{(t+1)}_{\text{reg}}$ to align it with $\mathbf{F}^{(t)}$, using the estimated flow $\mathbf{W}^{(t)}$. We apply the warp with a differentiable 3D spatial transformer layer, which does trilinear interpolation to resample each voxel. This extends self-supervised 2D optical flow (Yu et al., 2016) to 3D feature constancy (instead of 2D brightness constancy).

We found that both types of supervision are essential for obtaining accurate 3D flow field estimates. Since we are not interested in the 3D motion of empty air voxels, we additionally estimate 3D voxel occupancy, and supervise this using the input pointclouds. We describe our 3D occupancy estimation in more detail in the appendix. At test time, we set the 3D motion of all unoccupied voxels to zero.

The proposed 3D imagination flow enjoys significant benefits over 2D optical flow or 3D scene flow. It does not suffer from occlusions and dis-occlusions of image content or projection artifacts (Sun et al., 2010), which typically transform rigid 3D transformations into non-rigid 2D flow fields. In comparison to 3D scene flow (Hornacek et al., 2014), which concerns visible 3D points, 3D imagination flow is computed between visual features that may never have appeared in the field of view, but are rather filled in by the model (i.e., "imagined").

### 3.3.2 3D Moving Object Segmentation

We obtain 3D object segmentation proposals by thresholding the 3D imagination flow magnitude, and clustering voxels using connected components. We score each component using a 3D version of a center-surround motion saliency score employed by numerous works for 2D motion saliency detection (Gao et al., 2008; Mahadevan & Vasconcelos, 2010). This score is high when the 3D box interior has lots of motion but the surrounding shell does not. This results in a set of scored 3D segmentation proposals for each video scene.

## 4 Experiments

We train our models in CARLA (Dosovitskiy et al., 2017), an open-source photorealistic simulator of urban driving scenes, which permits moving the camera to any desired viewpoint in the scene. We obtain data from the simulator as follows. We generate 1170 autopilot episodes of 50 frames each (at 30 FPS), spanning all weather conditions and all locations in both "towns" in the simulator. We define 36 viewpoints placed regularly along a 20m-radius hemisphere in front of the ego-car. This hemisphere is anchored to the ego-car (i.e., it moves with the car). In each episode, we sample 6 random viewpoints from the 36 and randomly perturb their pose, and then capture each timestep of the episode from these 6 viewpoints. We generate train/test examples from this, by assembling all combinations of viewpoints (e.g., $N \leq 5$ viewpoints as input, and 1 unseen viewpoint as the target). We filter out frames that have zero objects within the metric "in bounds" region of the GRNN ($32m \times 32m \times 4m$). This yields 172524 frames (each with multiple views): 124256 in Town1, and 48268 in Town2. We treat the Town1 data as the "training" set, and the Town2 data as the "test" set, so there is no overlap between the train and test images.

For additional testing with real-world data, we use the (single-view) object detection benchmark from the KITTI dataset (Geiger et al., 2013), with the official train/val split: 3712 training frames, and 3769 validation frames.

We evaluate our view-contrastive 3D feature representations in three tasks: **(1)** semi-supervised 3D object detection, **(2)** unsupervised 3D moving object detection, and **(3)** 3D motion estimation. We use Tung et al. (2019) as a baseline representing the state-of-the-art, but evaluate additional related works in the appendix (see Sec. C.2, Sec. C.3).

## 4.1 SEMI-SUPERVISED LEARNING OF 3D OBJECT DETECTION

We use the proposed view-contrastive prediction as pre-training for 3D object detection [2]. We pretrain the inverse graphics network weights, and then train a 3D object detector module supervised to map a 3D feature volume **M** to 3D object boxes, as described in Section 3.1. We are interested in seeing the benefit of this pre-training across different amounts of label supervision, so we first use the full CARLA train set for view prediction training (without using box labels), and then use a randomly-sampled subset of the CARLA train set for box supervision; we evaluate on the CARLA validation set. We varied the size of the box supervision subset across the following range: 100, 200, 500, 1000, 10000, 80000. We show mean average precision (at an IoU of 0.75) for car detection as a function of the number of annotated 3D bounding box examples in Figure 1. We compare our model against a version of our model that optimizes for RGB regression, similar to Tung et al. (2019) but with a 6 DoF camera motion as opposed to 2 DoF, as well as a model trained from random weight initialization (i.e., without pre-training). After pre-training, we *freeze* the feature layers after view predictive learning, and only supervise the detector module; for the fully supervised baseline (from random initialization), we train end-to-end.

As expected, the supervised model performs better with more labelled data. In the low-data regime, pre-training greatly improves results, and more so for view-contrastive learning than RGB learning. We could not compare against alternative view prediction models as the overwhelming majority of them consider pre-segmented scenes (single object setups; e.g., Kar et al., 2017) and cannot generalize beyond those settings. The same is the case for the model of Eslami et al. (2018), which was greatly outperformed by GRNNs in the work of Tung et al. (2019).

### 4.1.1 SIM-TO-REAL (CARLA-TO-KITTI) TRANSFER

We evaluate whether the 3D predictive feature representations learned in the CARLA simulator are useful for learning 3D object detectors in the real world by testing on the real KITTI dataset (Geiger et al., 2013). Specifically, we use view prediction pre-training in the CARLA train set, and box supervision from the KITTI train set, and evaluate 3D object detection in the KITTI validation set.

Existing real-world datasets do not provide enough camera viewpoints to support view-predictive learning. Specifically, in KITTI, all the image sequences come from a moving car and thus all viewpoints lie on a near-straight trajectory. Thus, simulation-to-real transferability of features is especially important for view predictive learning.

We show sim-to-real transfer results in Table 1. We compare the proposed view-contrastive prediction pre-training with view regression pre-training and random weight initialization (no pre-training). In all cases, we train a 3D object detector on the features, supervised using KITTI 3D box annotations. We also compare *freezing* versus *finetuning* the weights of the pretrained inverse graphics network. The results are consistent with the CARLA tests: view-contrastive

| Method | mAP@IOU | | |
|---|---|---|---|
| | 0.33 | 0.50 | 0.75 |
| No pre-training (random init.) | .59 | .52 | .17 |
| View regression pret., frozen | .64 | .54 | .15 |
| View regression pret., finetuned | .65 | .55 | .18 |
| View-contrastive pret., frozen | .67 | .58 | .15 |
| View-contrastive pret., finetuned | **.70** | **.60** | **.19** |

Table 1: **CARLA-to-KITTI feature transfer**. We train a 3D detector module on top of the inferred 3D feature maps **M** using KITTI 3D object box annotations

pre-training is best, view regression pre-training is second, and learning from annotations alone is worst. Note that depth in KITTI is acquired by a real LiDAR sensor, and therefore has lower density and more artifacts than CARLA, yet our model generalizes across this distribution shift.

---

[2]Prior work has evaluated self-supervised 2D deep features in this way, by re-purposing them for a 2D detection task, e.g., Doersch et al. (2015).

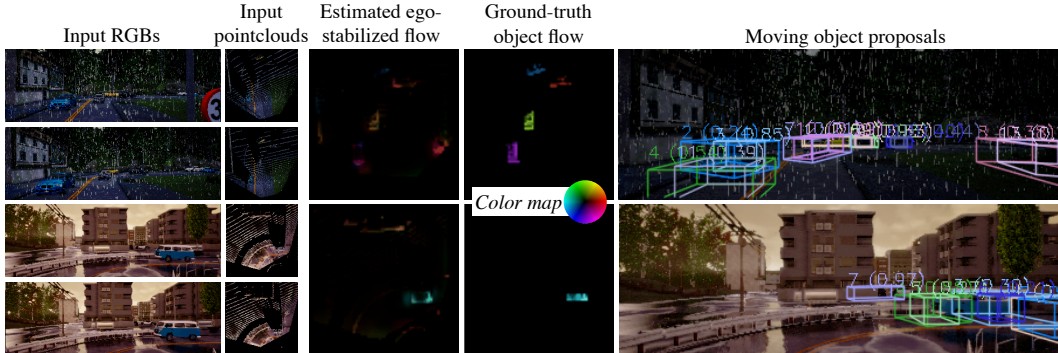

Figure 3: **3D feature flow and object proposals, in dynamic scenes.** Given the input frames on the left, our model estimates dense egomotion-stabilized 3D flow fields, and converts these into object proposals. We visualize colorized pointclouds and flow fields in a top-down (bird's eye) view.

## 4.2 UNSUPERVISED 3D MOVING OBJECT DETECTION

In this section, we test our model's ability to detect moving objects in 3D with no 3D object annotations, simply by clustering 3D motion vectors. We use two-frame sequences of dynamic scenes from our CARLA validation set, and split it into two parts for evaluation: scenes where the camera is stationary, and scenes where the camera is moving. This splitting is based on the observation that moving object detection is made substantially more challenging under a moving camera.

We show precision-recall curves for 3D moving object detection under a *stationary camera* in Figure 4. We compare our model against a similar one trained with RGB view regression (Tung et al., 2019) and a 2.5D baseline. The 2.5D baseline computes 2D optical flow using PWC-Net (Sun et al., 2018), then proposes object masks by thresholding and clustering 2D flow magnitudes; these 2D masks are mapped to 3D boxes by segmenting the input pointcloud and fitting boxes to the points. Our model outperforms the baselines. Note that even with ground-truth 2D flow, ground-truth depth, and an oracle threshold, a 2.5D baseline can at best only capture the currently-visible fragment of each object. As a result, a 3D proposal computed from 2.5D often underestimates the extent of the object. Our model does not have this issue, since it imagines the full 3D scene at each timestep.

We show precision-recall curves for 3D moving object detection under a *moving camera* in Figure 5. We compare our model where egomotion is predicted by our neural egomotion module, against our model with ground-truth egomotion, as well as a 2.5D baseline, and a stabilized 2.5D baseline. The 2.5D baseline uses optical flow estimated from PWC-Net as before. To stabilize the 2.5D flow, we subtract the ground-truth scene flow from the optical flow estimate before generating proposals. Our model's performance is similar to its level in static scenes, suggesting that the egomotion module's stabilization mechanism effectively disentangles camera motion from the 3D feature maps. The 2.5D baseline performs poorly in this setting, as expected. Surprisingly, performance drops further after stabilizing the 2D flows for egomotion. We confirmed this is due to the estimated scene flow being imperfect: subtracting ground-truth scene flow leaves many motion fragments in the background. With ground-truth 2D flow, the baseline performs similar to its static-scene level.

We have attempted to compare against the unsupervised object segmentation methods proposed in Kosiorek et al. (2018) and Hsieh et al. (2018) by adapting the publicly available code to the task. These models consume a video as input, and predict the locations of 2D object bounding boxes, as well as frame-to-frame displacements, in order to minimize a view regression error. We were not able to produce meaningful results from these models. The success of Hsieh et al. (2018) may partially depend on carefully selected priors for the 2D bounding box locations and sizes, to match the statistics of the "Moving MNIST" dataset used in that work (as suggested in the official code). For our CARLA experiments, we do not assume knowledge of priors for box locations or sizes.

## 4.3 3D MOTION ESTIMATION

In this section, we evaluate accuracy of our 3D imagination flow module. The previous section evaluated this module indirectly since it plays a large part in unsupervised 3D moving object detection;

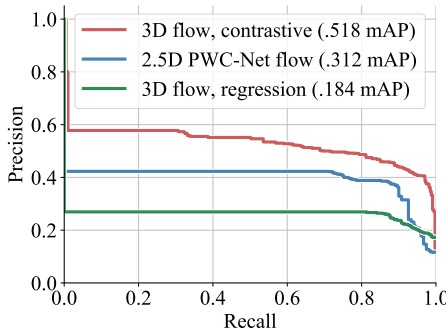
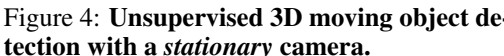

Figure 4: **Unsupervised 3D moving object detection with a *stationary* camera.**

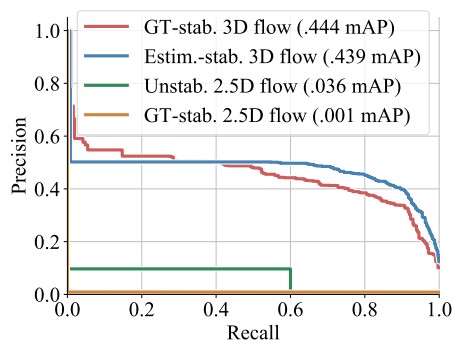

Figure 5: **Unsupervised 3D moving object detection with a *moving* camera**

here we evaluate its accuracy directly. We use two-frame video sequences of dynamic scenes from our CARLA validation set.

We compare 3D flow trained over frozen 3D feature representations obtained from the proposed view-contrastive prediction, against flow trained over frozen features from the baseline RGB regression model (Tung et al., 2019). We also compare against a zero-motion baseline that predicts zero

| Method | Full | Static | Moving |
|---|---|---|---|
| Zero-motion | **0.19** | **0.0** | 6.97 |
| View regression pret. | 0.77 | 0.63 | 5.55 |
| View-contrastive pret. | 0.26 | 0.17 | **3.33** |

Table 2: **Mean endpoint error of the 3D flow in egomotion-stabilized scenes.** View-contrastive features lead to more accurate motion estimation. Error is $L_1$ in voxel units.

motion everywhere. Since approximately 97% of the voxels belong to the static scene, a zero-motion baseline is very competitive in an overall average. We therefore evaluate error separately in static and moving parts of the scene. We show the motion estimation errors in Table 2. Our method achieves dramatically lower error than the RGB regression baseline, which suggests that the proposed view-contrastive objectives in 3D and 2D result in learning features that are correspondable across time, even for moving objects, despite the fact that the features were learned using only multi-view data of static scenes.

## 4.4 LIMITATIONS

The proposed model has two important limitations. First, our work assumes an embodied agent that can move around at will. This is hard to realize in the real world, and indeed there are almost no existing datasets with enough camera views to approximate this. Second, our model architecture consumes a lot of GPU memory, due to its extra spatial dimension . This severely limits either the resolution or the metric span of the latent map **M**. On 12G Titan X GPUs we encode a space sized $32m \times 32m \times 8m$ at a resolution of $128 \times 128 \times 32$; with a batch size of 4, iteration time is 0.2s/iter. Supervised 3D object detectors typically cover twice this metric range. Sparsifying our feature grid, or using points instead of voxels, are clear areas for future work.

## 5 CONCLUSION

We propose models that learn space-aware 3D feature abstractions of the world given 2.5D input, by minimizing 3D and 2D view-contrastive prediction objectives. We show that view-contrastive prediction leads to features useful for 3D object detection, both in simulation and in the real world. We further show that the ability to visually imagine full 3D egomotion-stable scenes allows us to estimate dense 3D motion fields, where clustering non-zero motion allows objects to emerge without any human supervision. Our experiments suggest that the ability to imagine 3D visual information can drive 3D object detection. Instead of learning from annotations, the model learns by moving and watching objects move (Gibson, 1979).

ACKNOWLEDGEMENTS

We thank Christopher G. Atkeson for providing the historical context of the debate on the utility and role of geometric 3D models versus feature-based models, and for many other helpful discussions.

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

## A  APPENDIX OVERVIEW

In Section B, we provide implementation details for our 3D-bottlenecked architecture, egomotion module, and 3D imagination flow module. In Section C, we provide additional experiments, and additional visualizations of our output. In Section D, we discuss additional related work.

## B  ARCHITECTURE DETAILS

**Inputs**  Our input images are sized $128 \times 384$ pixels. We trim the input pointclouds to a maximum of 100,000 points, and to a range of 80 meters, to simulate a Velodyne LiDAR sensor.

**2D-to-3D unprojection**  This module converts the input 2D image $I^{(t)}$ and pointcloud $D^{(t)}$ into a 4D tensor $\mathbf{U}^{(t)} \in \mathbb{R}^{w \times h \times d \times 4}$, by filling the 3D imagination grid with samples from the 2D image grid, using perspective (un)projection. Specifically, for each cell in the imagination grid, indexed by the coordinate $(i, j, k)$, we compute the floating-point 2D pixel location $[u, v]^T = \boldsymbol{KS}[i, j, k]^T$ that it projects to from the current camera viewpoint, using the pinhole camera model (Hartley & Zisserman, 2003), where $\boldsymbol{S}$ is the similarity transform that converts memory coordinates to camera coordinates and $\boldsymbol{K}$ is the camera intrinsics (transforming camera coordinates to pixel coordinates). We fill $\mathbf{U}^{(t)}_{i,j,k}$ with the bilinearly interpolated pixel value $I^{(t)}_{u,v}$. We transform our depth map $D_t$ in a similar way and obtain a binary occupancy grid $\mathbf{O}^{(t)} \in \mathbb{R}^{w \times h \times d \times 1}$, by assigning each voxel a value of 1 or 0, depending on whether or not a point lands in the voxel. We concatenate this to the unprojected RGB, making the tensor $[\mathbf{U}^{(t)}, \mathbf{O}^{(t)}] \in \mathbb{R}^{w \times h \times d \times 4}$.

We pass the tensors $[\mathbf{U}^{(t)}, \mathbf{O}^{(t)}]$ through a 3D encoder-decoder network. The 3D feature encoder-decoder has the following architecture, using the notation $k$-$s$-$c$ for kernel-stride-channels: 4-2-64, 4-2-128, 4-2-256, 4-0.5-128, 4-0.5-64, 1-1-$F$, where $F$ is the feature dimension. We use $F = 32$. After each transposed convolution in the decoder, we concatenate the same-resolution featuremap from the encoder. Every convolution layer (except the last in each net) is followed by a leaky ReLU activation and batch normalization.

**Egomotion estimation**  This module computes the relative 3D rotation and translation between the current camera viewpoint and the reference coordinate system of the map $\mathbf{M}^{(1)}$. We significantly changed the module of Tung et al. (2019) which could only handle 2 degrees of camera motion. We consider a general camera with full 6-DoF motion. Our egomotion module is inspired by the state-of-the-art PWC-Net optical flow method (Sun et al., 2018): it incorporates spatial pyramids, incremental warping, and cost volume estimation via cross-correlation.

While $\mathbf{M}^{(1)}$ and $\mathbf{M}^{(t)}$ can be used directly as input to the egomotion module, we find better performance can be obtained by allowing the egomotion module to learn its own feature space. Thus, we begin by passing the (unregistered) 3D inputs through a 3D encoder-decoder, producing a reference tensor $\mathbf{F}^{(1)} \in \mathbb{R}^{w \times h \times d \times c}$, and a query tensor $\mathbf{F}^{(t)} \in \mathbb{R}^{w \times h \times d \times c}$. We wish to find the rigid transformation that aligns the two.

We use a coarse-to-fine architecture, which estimates a coarse 6D answer at the coarse scale, and refines this answer in a finer scale. We iterate across scales in the following manner: First, we downsample both feature tensors to the target scale (unless we are at the finest scale). Then, we generate several 3D rotations of the second tensor, representing "candidate rotations", making a set $\{\mathbf{F}^{(t)}_{\theta_i} | \theta_i \in \Theta\}$, where $\Theta$ is the discrete set of 3D rotations considered. We then use 3D axis-aligned cross-correlations between $\mathbf{F}^{(1)}$ and the $\mathbf{F}^{(t)}_{\theta_i}$, which yields a cost volume of shape $r \times w \times h \times d \times e$, where $e$ is the total number of spatial positions explored by cross-correlation. We average across spatial dimensions, yielding a tensor shaped $r \times e$, representing an average alignment score for each transform. We then apply a small fully-connected network to convert these scores into a 6D vector. We then warp $\mathbf{F}^{(t)}$ according to the rigid transform specified by the 6D vector, to bring it into (closer) alignment with $\mathbf{F}^{(1)}$. We repeat this process at each scale, accumulating increasingly fine corrections to the initial 6D vector.

Similar to PWC-Net (Sun et al., 2018), since we compute egomotion in a coarse-to-fine manner, we need only consider a small set of rotations and translations at each scale (when generating the

cost volumes); the final transform composes all incremental transforms together. However, unlike PWC-Net, we do not repeatedly warp our input tensors, because this accumulates interpolation error. Instead, we follow the inverse compositional Lucas-Kanade algorithm (Baker & Matthews, 2004; Lin & Lucey, 2017), and at each scale warp the original input tensor with the composed transform.

**3D-to-2D projection** This module "renders" 2D feature maps given a desired viewpoint $\boldsymbol{V}^{(k)}$ by projecting the 3D feature state $\mathbf{M}^{(t)}$. We first appropriately orient the 3D featuremap by re-sampling $\mathbf{M}^{(t)}$ into a view-aligned version $\mathbf{M}^{(t)}_{\text{view}_k}$. The sampling is defined by $[i', j', k']^T = \boldsymbol{S}^{-1}\boldsymbol{V}^{(k)}\boldsymbol{S}[i, j, k]^T$, where $\boldsymbol{S}$ is (as before) the similarity transform that brings imagination co-ordinates into camera coordinates, $\boldsymbol{V}^{(k)}$ is the transformation that relates the reference camera coordinates to the viewing camera coordinates, $[i, j, k]$ are voxel indices in $\mathbf{M}^{(t)}$, and $[i', j', k']$ are voxel indices in $\mathbf{M}^{(t)}_{\text{view}_k}$. We then warp the view-oriented tensor $\mathbf{M}^{(t)}_{\text{view}_k}$ such that perspective viewing rays become axis-aligned. We implement this by sampling from the memory tensor with the correspondence $[u, v, d]^T = \boldsymbol{K}\boldsymbol{S}[i', j', k']^T$, where the indices $[u, v]$ span the image we wish to generate, and $d$ spans the length of each ray. We use logarithmic spacing for the increments of $d$, finding it far more effective than linear spacing (used in Tung et al., 2019), likely because our scenes cover a large metric space. We call the perspective-transformed tensor $\mathbf{M}^{(t)}_{\text{proj}_k}$. To avoid repeated interpolation, we actually compose the view transform with the perspective transform, and compute $\mathbf{M}^{(t)}_{\text{proj}_k}$ from $\mathbf{M}^{(t)}$ with a single trilinear sampling step. Finally, we pass the perspective-transformed tensor through a CNN, converting it to a 2D feature map $M^{(t)}_{\text{view}_k}$. The CNN has the following architecture (using the notation $k$-$s$-$c$ for kernel-stride-channels): max-pool along the depth axis with $1 \times 8 \times 1$ kernel and $1 \times 8 \times 1$ stride, to coarsely aggregate along each camera ray, 3D convolution with 3-1-32, reshape to place rays together with the channel axis, 2D convolution with 3-1-32, and finally 2D convolution with 1-1-$E$, where $E$ is the channel dimension. For predicting RGB, $E = 3$; for metric learning, we use $E = 32$. We find that with dimensionality $E = 16$ or less, the model underfits.

**3D imagination flow** To train our 3D flow module, we generate supervised labels from synthetic transformations of single-timestep input, and an unsupervised loss based on the standard standard variational loss (Horn & Schunck, 1981; Yu et al., 2016). For the synthetic transformations, we randomly sample from three uniform distributions of rigid transformations: (i) *large motion*, with rotation angles in the range $[-6, 6]$ (degrees) and translations in $[-1, 1]$ (meters), (ii) *small motion*, with angles from $[-1, 1]$ and translations from $[-0.1, 0.1]$, (iii) *zero motion*. We found that without sampling (additional) small and zero motions, the model does not accurately learn these ranges. Still, since these synthetic transformations cause the entire tensor to move at once, a flow module learned from this supervision alone tends to produce overly-smooth flow in scenes with real (non-rigid) motion. The variational loss $\mathcal{L}_{\text{warp}}$, described in the main text, overcomes this issue.

We also apply a smoothness loss penalizing local flow changes: $\mathcal{L}_{\text{smooth}} = \sum_{i,j,k} ||\nabla \mathbf{W}^{(t)}{}_{i,j,k}||$, where $\mathbf{W}^{(t)}$ is the estimated flow field and $\nabla$ is the 3D spatial gradient. This is a standard technique to prevent the model from only learning motion edges (Horn & Schunck, 1981; Yu et al., 2016).

**3D occupancy estimation** The goal in this step is to estimate which voxels in the imagination grid are "occupied" (i.e., have something visible inside) and which are "free" (i.e., have nothing visible inside). For supervision, we obtain (partial) labels for both "free" and "occupied" voxels using the input depth data. Sparse "occupied" voxel labels are given by the voxelized pointcloud $\mathbf{O}^{(t)}_{\text{reg}}$. To obtain labels of "free" voxels, we trace the source-camera ray to each occupied observed voxel, and mark all voxels intersected by this ray as "free".

Our occupancy module takes the memory tensor $\mathbf{M}^{(t)}$ as input, and produces a new tensor $\mathbf{P}^{(t)}$, with a value in $[0, 1]$ at each voxel, representing the probability of the voxel being occupied. This is achieved by a single 3D convolution layer with a $1 \times 1 \times 1$ filter (or, equivalently, a fully-connected network applied at each grid location), followed by a sigmoid nonlinearity. We train this network with the logistic loss, $\mathcal{L}_{\text{occ}} = (1/\sum \hat{\mathbf{I}}^{(t)}) \sum \hat{\mathbf{I}}^{(t)} \log(1 + \exp(-\hat{\mathbf{P}}^{(t)}\mathbf{P}^{(t)}))$, where $\hat{\mathbf{P}}^{(t)}$ is the label map, and $\hat{\mathbf{I}}^{(t)}$ is an indicator tensor, indicating which labels are valid. Since there are far more "free" voxels than "occupied", we balance this loss across classes within each minibatch.

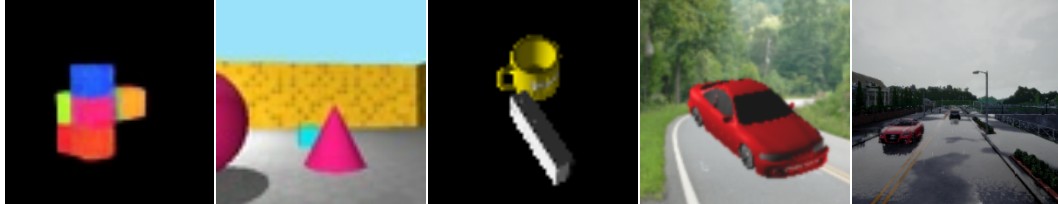

Figure 6: **Dataset comparison for view prediction.** From left to right: Shepard-Metzler (Eslami et al., 2018), Rooms-Ring-Camera (Eslami et al., 2018), ShapeNet arrangements (Tung et al., 2019), cars (Tatarchenko et al., 2016) and CARLA (used in this work) (Dosovitskiy et al., 2017). CARLA scenes are more realistic, and are not object-centric.

**2D CNN** The CNN that converts the target view into an embedding image is a residual network (He et al., 2016) with two residual blocks, with 3 convolutions in each block. The convolution layers' channel dimensions are 64, 64, 64, 128, 128, 128. Finally there is one convolution layer with $E$ channels, where $E$ is the embedding dimension. We use $E = 32$.

**Contrastive loss** For both the 2D and the 3D contrastive loss, for each example in the minibatch, we randomly sample a set of 960 pixel/voxel coordinates $\mathcal{S}$ for supervision. Each coordinate $i \in \mathcal{S}$ gives a positive correspondence $\mathcal{T}_i, \mathcal{B}_i$, since the tensors are aligned. For each $\mathcal{T}_i$, we sample a negative $\mathcal{B}_k$ from the samples acquired across the entire batch, using the distance-weighted sampling strategy of Wu et al. (2017). In this way, on every iteration we obtain an equal number of positive and negative samples, where the negative samples are spread out in distance. We additionally apply an $L_2$ loss on the difference between the entire tensors, which penalizes distance at *all* positive correspondences (instead of merely the ones sampled for the metric loss). We find that this accelerates training. We use a coefficient of 0.1 for $\mathcal{L}^{\text{2D}}_{\text{contrast}}$, 1.0 for $\mathcal{L}^{\text{3D}}_{\text{contrast}}$, and 0.001 for the $L_2$ losses.

**Code and training details** Our model is implemented in Python/Tensorflow, with custom CUDA kernels for the 3D cross correlation (used in the egomotion module and the flow module) and for the trilinear resampling (used in the 2D-to-3D and 3D-to-2D modules). The CUDA operations use less memory than native-tensorflow equivalents, which facilitates training with large imagination tensors ($128 \times 128 \times 32 \times 32$). Training to convergence (approx. 200k iterations) takes 48 hours on a single GPU. We use a learning rate of 0.001 for all modules except the 3D flow module, for which we use 0.0001. We use the Adam optimizer, with $\beta_1 = 0.9$, $\beta_2 = 0.999$.

## C    ADDITIONAL EXPERIMENTS

### C.1    DATASETS

**CARLA vs. other datasets** We test our method on scenes we collected from the CARLA simulator (Dosovitskiy et al., 2017), an open-source driving simulator of urban scenes. CARLA permits moving the camera to any desired viewpoint in the scene, which is necessary for our view-based learning strategy. Previous view prediction works have considered highly synthetic datasets: The work of Eslami et al. (2018) introduced the Shepard-Metzler dataset, which consists of seven colored cubes stuck together in random arrangements, and the Rooms-Ring-Camera dataset, which consists of a random floor and wall colors and textures with variable numbers of shapes in each room of different geometries and colors. The work of Tung et al. (2019) introduced a ShapeNet arrangements dataset, which consists of table arrangements of ShapeNet synthetic models (Chang et al., 2015). The work of Tatarchenko et al. (2016) considers scenes with a single car. Such highly synthetic and limited-complexity datasets cast doubt on the usefulness and generality of view prediction for visual feature learning. The CARLA simulation environments considered in this work have photo-realistic rendering, and depict diverse weather conditions, shadows, and objects, and arguably are much closer to real world conditions, as shown in Figure 6. While there exist real-world datasets which are visually similar (Geiger et al., 2013; Caesar et al., 2019), they only contain a small number viewpoints, which makes view-predictive training inapplicable.

Since occlusion is a major factor in a dataset's difficulty, we provide occlusion statistics collected from our CARLA data. Note that in a 2D or unrealistic 3D world, most of the scene would be fully

| Method | 2D patch retrieval | | | 3D patch retrieval | | |
|---|---|---|---|---|---|---|
| | P@1 | P@5 | P@10 | P@1 | P@5 | P@10 |
| **2D-bottlenecked regression architectures** | | | | | | |
| GQN with 2D L1 loss (Eslami et al., 2018) | .00 | .01 | .02 | - | - | - |
| **3D-bottlenecked regression architectures** | | | | | | |
| GRNN with 2D L1 loss (Tung et al., 2019) | .00 | .00 | .00 | .42 | .57 | .60 |
| GRNN-VAE with 2D L1 loss & KL div. | .00 | .00 | .01 | .34 | .58 | .66 |
| **3D bottlenecked contrastive architectures** | | | | | | |
| GRNN with 2D contrastive loss | .14 | .29 | .39 | .52 | .73 | .79 |
| GRNN with 2D & 3D contrastive losses | **.20** | **.39** | **.47** | **.80** | **.97** | **.99** |

Table 3: Retrieval precision at different recall thresholds, in 2D and in 3D. P@$K$ indicates precision at the $K$th rank. Contrastive learning outperforms regression learning by a substantial margin.

| Input View | Target View | GRNN | VAE-GRNN | GQN |
|---|---|---|---|---|

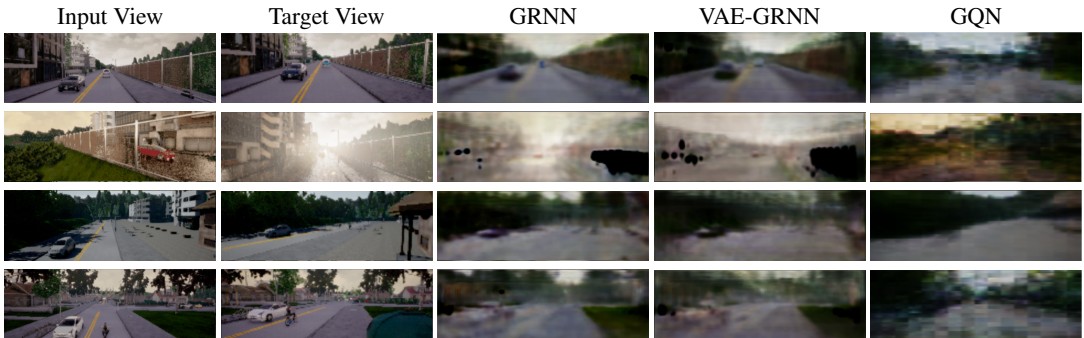

Figure 7: **View prediction in CARLA.** For each input, we show the target view and view prediction attempts, by a GRNN-style architecture (Tung et al., 2019), a VAE variant of it, and GQN (Eslami et al., 2018). The predictions are blurry and only coarsely correct, due to the complexity of the RGB.

visible in every image. In CARLA, a single camera view reveals information on approximately $0.23 (\pm 0.03)$ of all voxels in the model's $32m \times 32m \times 8m$ "in bounds" space, leaving 0.77 totally occluded/unobserved. This measure includes both "scene surface" voxels, and voxels intersecting rays that travel from the camera center to the scene surface (i.e., "known free space"). The surface itself occupies only $0.01 (\pm 0.002)$ of the volume. Adding a random second view, the total volume revealed is $0.28 (\pm 0.05)$; the surface revealed is $0.02 (\pm 0.003)$. With all 6 views, $0.42 (\pm 0.04)$ of the volume is revealed; $0.03 (\pm 0.004)$ is revealed surface. These statistics illustrate that the vast majority of the scene must be "imagined" by the model to satisfy the contrastive prediction objectives.

## C.2 RGB VIEW PREDICTION

Images from the CARLA simulator have complex textures and specularities and are close to photorealism, which causes RGB view prediction methods to fail. We illustrate this in Figure 7: given an input image and target viewpoint (i.e., pose), we show target views predicted by a neural 3D mapper trained for RGB prediction (Tung et al., 2019), (ii) a VAE variant of that architecture, and (iii) Generative Query Networks (GQN; Eslami et al., 2018, which does not have a 3D representation bottleneck, but rather concatenates 2D images and their poses in a 2D recurrent architecture. Unlike these works, our model does not use view prediction as the end task, but rather as a means of learning useful visual representation for 3D object detection, segmentation and motion estimation.

## C.3 2D AND 3D CORRESPONDENCE

We evaluate our model's performance in estimating visual correspondences in 2D and in 3D, using a nearest-neighbor retrieval task. In 2D, the task is as follows: we extract one "query" patch from a top-down render of a viewpoint, then extract 1000 candidate patches from bottom-up renders, with only one true correspondence (i.e., 999 negatives). We then rank all bottom-up patches according to

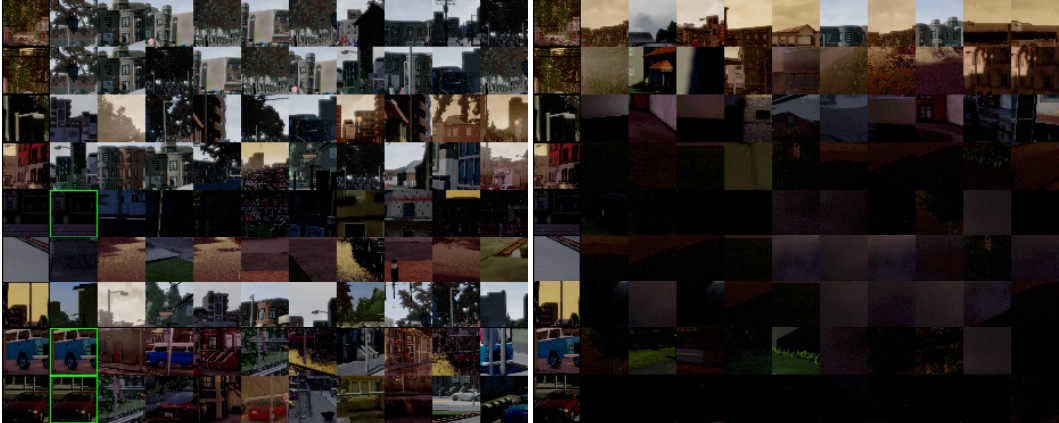

Figure 8: **2D image patch retrievals** acquired with the contrastive model (left) vs the regression model (right). Each row corresponds to a query. For each model, the query is shown on the far left, and the 10 nearest neighbors are shown in ascending order of distance. Correct retrievals are highlighted with a green border. The neighbors of the contrastive model often have clear semantic relationships (e.g., curbs, windows); the neighbors of the RGB model do not.

$L_2$ distance from the query, and report the retrieval precision at several recall thresholds, averaged over 1000 queries. In 3D the task is similar, but patches are *feature cubes* extracted from the 3D imagination; we generate queries from one viewpoint, and retrievals from other viewpoints and other scenes. The 1000 samples are generated as follows: from 100 random test examples, we generate 10 samples from each, so that each sample has 9 negatives from the same viewpoint, and 990 others from different locations/scenes.

We compare the proposed model against (i) the RGB prediction baseline of Tung et al. (2019), (ii) Generative Query Networks (GQN) of Eslami et al. (2018), which do not have a 3D representation bottleneck, and (iii) a VAE alternative of the (deterministic) model of Tung et al. (2019).

Quantitative results are shown in Table 3. For 2D correspondence, the models learned through the RGB prediction objectives obtain precision near zero at each recall threshold, illustrating that the model is not learning precise RGB predictions. The proposed view-contrastive losses perform better, and combining both the 2D and 3D contrastive losses is better than using only 2D. Interestingly, for 3D correspondence, the retrieval accuracy of the RGB-based models is relatively high. Training neural 3D mappers as variational autoencoders, where stochasticity is added in the 3D bottleneck, improves its precision at lower ranks thresholds. Contrastive learning outperforms all baselines. Adding the 3D contrastive loss gives a large boost over using the 2D contrastive loss alone. Note that 2D-bottlenecked architectures (Eslami et al., 2018) cannot perform 3D patch retrieval. Qualitative retrieval results for our full model vs. Tung et al. (2019) are shown in Figure 8.

### C.4 Unsupervised 3D Object Motion Segmentation

Our method proposes 3D object segmentations, but labels are only available in the form of oriented 3D boxes; we therefore convert our segmentations into boxes by fitting minimum-volume oriented 3D boxes to the segmentations. The precision-recall curves presented in the paper are computed with an intersection-over-union (IOU) threshold of 0.5. Figure 9 shows sample visualizations of 3D box proposals projected onto input images.

### C.5 Occupancy Prediction

We test our model's ability to estimate occupied and free space. Given a single view as input, the model outputs an occupancy probability for each voxel in the scene. Then, given the aggregated labels computed from this view and a random next view, we compute accuracy at all voxels for which we have labels. Voxels that are not intersected by either view's camera rays are left unlabelled. Table 4 shows the classification accuracy, evaluated independently for free and occupied voxels, and

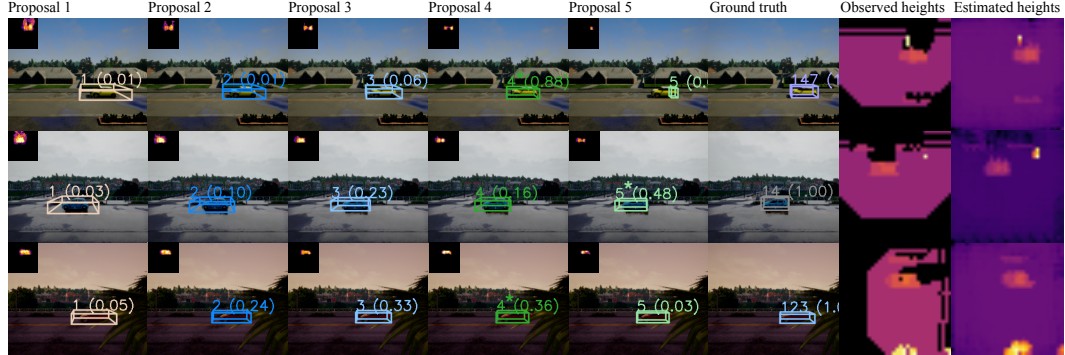

Figure 9: **Left: 3D object proposals and their center-surround scores** (normalized to the range [0,1]). For each proposal, the inset displays the corresponding connected component of the 3D flow field, viewed from a top-down perspective. In each row, an asterisk marks the box with the highest center-surround score. **Right: Observed and estimated heightmaps** of the given frames, computed from 3D occupancy grids. Note that the observed (ground truth) heightmaps have view-dependent "shadows" due to occlusions, while the estimated heightmaps are dense and viewpoint-invariant.

for all voxels aggregated together. Overall, accuracy is extremely high (97-98%) for both voxel types. Note that part of the occupied space (i.e., the voxelized pointcloud of the first frame) is an input to the network, so accuracy on this metric is expected to be high.

We show a visualization of the estimated occupancy volumes in Figure 9-right. We visualize the occupancy volumes by converting them to heightmaps. This is achieved by multiplying each voxel's occupancy value by its height coordinate in the grid, and then taking a max along the grid's height axis. The visualizations show that the occupancy module learns to fill the "holes" of the partial view.

| 3D area type | Accuracy |
|---|---|
| Occupied space | .97 |
| Free space | .98 |
| All space | .98 |

Table 4: **Occupancy classification accuracy.** Accuracy is nearly perfect in each region type.

## C.6    Egomotion Estimation

We compare our egomotion module against ORB-SLAM2 (Mur-Artal & Tardós, 2017), a state-of-the-art geometric simultaneous localization and mapping (SLAM) method, and against the SfM-Net (Zhou et al.,

| Method | $R$ (rad) | $t$ (m) |
|---|---|---|
| ORB-SLAM2 (Mur-Artal & Tardós, 2017) | 0.089 | 0.038 |
| SfM-Net (Zhou et al., 2017) | **0.083** | 0.086 |
| SfM-Net + GT depth | 0.100 | 0.078 |
| Ours | 0.120 | **0.036** |

Table 5: **Egomotion error.** Our model is on par with the baselines.

2017) architecture, which is a 2D CNN that takes pairs of frames and outputs egomotion. We give ORB-SLAM2 access to ground-truth pointclouds, but note that it is being deployed merely as an egomotion module (rather than for SLAM). We ran our own model and SfM-Net with images sized $128 \times 384$, but found that ORB-SLAM2 performs best at $256 \times 768$. SfM-Net is designed to estimate depth and egomotion unsupervised, but since our egomotion module is supervised, we supervise SfM-Net here as well. We evaluate two versions of it: one with RGB inputs (as designed), and one with RGB-D inputs (more similar to our model). Table 5 shows the results. Overall the models all perform similarly, suggesting that our egomotion method performs on par with the rest.

Note that the egomotion module of Tung et al. (2019) is inapplicable to this task, since it assumes that the camera orbits about a fixed point, with 2 degrees of freedom. Here, the camera is free, with 6 degrees of freedom.

## D ADDITIONAL RELATED WORK

**3D feature representations**   A long-standing debate in Computer Vision is whether it is worth pursuing 3D models in the form of binary voxel grids, meshes, or 3D pointclouds as the output of visual recognition. The "blocks world" of Roberts (1965) set as its goal to reconstruct the 3D scene depicted in the image in terms of 3D solids found in a database. Pointing out that replicating the 3D world in one's head is not enough to actually make decisions, Brooks (1991) argued for feature-based representations, as done by recent work in end-to-end deep learning (Levine et al., 2016). Our work proposes *learning-based 3D feature representations* in place of previous human-engineered ones, attempting to reconcile the two sides of the debate.

Some recent works have attempted various forms of map-building (Gupta et al., 2017; Henriques & Vedaldi, 2018) as a form of geometrically-consistent temporal integration of visual information, in place of geometry-unaware long short-term memory (LSTM; Hochreiter & Schmidhuber, 1997) or gated recurrent unit (GRU; Cho et al., 2014) models. The closest to ours are Learned Stereo Machines (LSMs; Kar et al., 2017), DeepVoxels (Sitzmann et al., 2019), and Geometry-aware Recurrent Neural Nets (GRNNs; Tung et al., 2019), which integrate images sampled from a viewing sphere into a latent 3D feature memory tensor, in an egomotion-stabilized manner, and predict views. All of these works consider very simplistic non-photorealistic environments and 2-degree-of-freedom cameras (i.e., with the camera locked to a pre-defined viewing sphere). None of these works evaluate the suitability of their learned features for a downstream task. Rather, their main objective is to accurately predict views.

**Motion estimation and object segmentation**   Most motion segmentation methods operate in 2D image space, and use 2D optical flow to segment moving objects. While earlier approaches attempted motion segmentation completely unsupervised, by exploiting motion trajectories and integrating motion information over time (Brox & Malik, 2010; Ochs & Brox, 2011), recent works focus on learning to segment objects in videos, supervised by annotated video benchmarks (Fragkiadaki et al., 2015; Khoreva et al., 2017). Our work differs in the fact that we address object detection and segmentation in 3D as opposed to 2D, by estimating 3D motion of the "imagined" (complete) scene, as opposed to 2D motion of the observed scene.

