# OpenReview forum: "Learning from Unlabelled Videos Using Contrastive Predictive Neural 3D Mapping"
_ICLR.cc/2020/Conference — Accept (Poster)_

### Official Review · AnonReviewer1 · 2019-10-26
**Official Blind Review #1**

**Rating:** 6

**Review:**

This paper studies the problem of visual representation learning from 2.5D video streams by exploring the 2D-3D geometry structures in the 3D visual world. Building upon the previous work GRNN (Tung et al. 2019), this paper introduced a novel view-contrast objective applied to its internal 2D and 3D feature space. To facilitate the 3D view-contrast learning, this paper proposed a novel 2D-3D inverse graphics networks with a 2D-to-3D un-projection encoder, a 2D encoder, a 3D bottlenecked RNNs, an ego-motion stabilization module, and a 3D-to-2D projection module. Compared to previous work (Tung et al. 2019), view-contrastive inverse graphics networks decode in the feature space rather than RGB space. Experimental evaluations are conducted using CARLA simulator (sim) and KITTI dataset (real). Results demonstrate the strengths of the proposed view-contrastive framework in feature learning, 3D moving object detection, and 3D motion estimation.

Overall, this paper studies an important problem in computer vision with a novel solution using unsupervised feature learning. While the technical novelty is clear, reviewer has several questions regarding the implementation and experimental details.

(1) For 3D box detection on KITTI (see Table 1), the comparisons to state-of-the-art models are currently missing. While the benefit of unsupervised feature learning has been demonstrated, it would be more convincing to compare against the following papers (at least with a paragraph of discussion).

(2) The 3D-to-2D projection module seems very expensive. Can you possibly report the training and inference time compared to baselines? Also, the design of the projection module is a bit counter-intuitive as it has 8x8 convolutions. In principle, such projection should be learning-free or with only 1x1 convolutions (aggregation along depth channel). It would be good to consider such ablation studies in the final version.

-- Multi-view Supervision for Single-view Reconstruction via Differentiable Ray Consistency. Tulsiani et al. In CVPR 2017.
-- Perspective Transformer Nets: Learning Single-View 3D Object Reconstruction without 3D Supervision. Yan et al. In NIPS 2016.
-- MarrNet: 3D Shape Reconstruction via 2.5D Sketches. Wu et al. In NIPS 2017.

(3) It seems that the proposed method assumes slow moving background across consecutive frames. In principle, the view-contrastive objective should mask out new pixels in frame T+1. Also, because the view-contrastive loss is applied at feature-level, reviewer would like to know performance on detecting small objects.

(4) As the latent map update module uses an RNN, it would be good to consider consistency beyond 2 frames (given mask is applied to view-contrastive objective). Curriculum learning could be helpful for further improvements.

-- Weakly-supervised Disentangling with Recurrent Transformations for 3D View Synthesis. Yang et al. In NIPS 2015.

(5) How does the proposed method perform when applied to indoor environments?

(6) Additional ablation study to consider: what if 2D/3D contrastive loss is turned off?



**Experience Assessment:**

I have published in this field for several years.

**Review Assessment: Checking Correctness Of Derivations And Theory:**

I carefully checked the derivations and theory.

**Review Assessment: Checking Correctness Of Experiments:**

I carefully checked the experiments.

**Review Assessment: Thoroughness In Paper Reading:**

I read the paper thoroughly.

---

> ### Author Response · Authors · 2019-11-12
> **Response to review (part 1/2)**
>
> Dear reviewer,
>
> Thank you for your detailed analysis.
>
> -- For 3D box detection on KITTI (see Table 1), the comparisons to state-of-the-art models are currently missing. While the benefit of unsupervised feature learning has been demonstrated, it would be more convincing to compare against the following papers.
>
> The state-of-the-art for KITTI is at approximately 0.8 mAP at the 0.5 IOU threshold, which exceeds our 0.6 mAP score. The main reason for this is resolution. The SOTA models in KITTI achieve extremely high resolution outputs, by sacrificing the “vertical” dimension of their latent representation: they build a “bird’s eye view” of the scene, and do 2D convolutions in that space. Our approach aims to be more general, and treats all three axes equally; we therefore use 3D convolutions throughout our model. Nonetheless, we are actively trying to bridge this resolution gap, via efficient implementations of sparse 3D convolutions: by avoiding wasteful computation on the “empty air” part of the scene, we are able to nearly double our resolution. Competing with SOTA on KITTI also requires careful augmentation of the pointclouds, via object-centric and scene-centric jittering and non-rigid transforms, to expand the training distribution to cover test statistics. We are doing this also, and hope to have results soon, but please note that higher resolution and augmented data lead to slower training, so it will take several days for each model to train. With luck, we will add these results into the paper by Friday.
>
>
> -- The 3D-to-2D projection module seems very expensive. Can you possibly report the training and inference time compared to baselines? Also, the design of the projection module is a bit counter-intuitive as it has 8x8 convolutions. In principle, such projection should be learning-free or with only 1x1 convolutions (aggregation along depth channel). It would be good to consider such ablation studies in the final version.
>
> Thank you for writing this. Our description should have been better. Our 3D-to-2D architecture is actually quite cheap. (There are no 8x8 convolutions.)
> - The module starts with a perspective transform, which puts viewing rays along the depth axis of a tensor.
> - Then there is a max pooling operation with a 1x8x1 kernel and 1x8x1 stride, where the 8 is along the depth axis; this quickly and coarsely aggregates along the ray axis, as you suggest.
> - Following this aggregation, there is a 3D convolution with kernel size 3x3x3.
> - Then there is a reshape, putting the depth dimension together with channels.
> - Finally there are two 2D convolution layers, with a 3x3 kernel then a 1x1 kernel.
> We have updated the text to clarify this.
>
> Regarding the related works you pointed out: Tulsiani et al. does not actually have an image renderer: that work only has a 2D-to-3D mapping, and then uses losses to encourage the 3D representation to be consistent with the 2.5D data (RGB-D images). Wu et al. is similar to Tulsiani et al.: they have no renderer and apply losses directly in 3D voxelspace, except they assume an orthographic camera instead of a perspective one. Our work’s “perspective transform” step is very similar to the first step in Yan et al., except they follow this by a single depthwise maxpool and output the result; this has no capacity for occlusion-reasoning and cannot be effective when the scene includes a background. The Yang et al. model uses a 1D (vector) latent space uses fully-connected layers for the transformations; our model’s 3D latent space allows for explicit geometric transformations. Another related work is DeepVoxels (Sitzmann et al.), which has a softmax on the ray axis, followed by a dot product. We tried this, and found it did not work better than our CNN-based renderer. We will add this result into the paper, as you suggest.
>
>
> -- It seems that the proposed method assumes slow moving background across consecutive frames.
>
> This is a great observation. We do not explicitly assume a slow-moving background, but the egomotion module (with its coarse-to-fine cross correlations) has a limited effective range. In particular, we use 3 scales in this module (0.25, 0.5, 1.0), and the correlations have a maximum displacement of 3 voxels; this means a limit of approximately 5 meters displacement between timesteps. The rotation limit is approximately 8 degrees. We chose these limits based on the statistics of camera motion in the KITTI dataset. You are right that if the camera motion becomes unexpectedly extreme at test time, we will not be able to “cancel it out” as desired, since our egomotion estimation will fail.
>
>
> (We continue in the next comment, due to the OpenReview character count limit.)

---

> > ### Author Response · Authors · 2019-11-12
> > **Response to review (part 2/2)**
> >
> > -- In principle, the view-contrastive objective should mask out new pixels in frame T+1.
> >
> > Masking out the new pixels would be appropriate for a view prediction model based on flow/warping, but that is actually not our setting. We are particularly interested in a model that is able to “imagine” or “inpaint” the full scene in its latent space, despite only having a partial observation of it. Therefore, the “new” pixels from T+1 are important to include in the final loss, since these can measure the model’s ability to fill in the missing information.
> >
> >
> > -- Because the view-contrastive loss is applied at feature-level, reviewer would like to know performance on detecting small objects.
> >
> > We agree, this would be an interesting evaluation, and may provide additional insights on the impact of our model’s relatively low resolution. We do not have this result yet, but we can stratify the data based on object size, and evaluate this.
> >
> >
> > -- As the latent map update module uses an RNN, it would be good to consider consistency beyond 2 frames (given mask is applied to view-contrastive objective). Curriculum learning could be helpful for further improvements.
> >
> > Thank you for these suggestions. In static scenes, (e.g., for semi-supervised object detection,) our model can integrate over an arbitrary number of frames. In dynamic scenes, we use just 2 frames for moving object discovery, while indeed a longer horizon may help. Achieving this long-term temporal consistency in feature space is an area for future work. We agree, curriculum learning may be an excellent approach here, since it would allow us to add gradually longer and longer consistency into the model.
> >
> >
> > -- How does the proposed method perform when applied to indoor environments?
> >
> > We have recently tried our model in the Habitat indoor scene simulator (Savva et al., ICCV 2019), and the method performs well, as expected. We use CARLA in this paper as it is the only realistic open-source simulator with dynamic (independently moving)  objects.
> >
> >
> > -- Additional ablation study to consider: what if 2D/3D contrastive loss is turned off?
> >
> > Yes, this was actually one of our preliminary experiments, and we have included a version of it in the appendix, in Table 3 (Sec. C.3). In particular, we measured the multi-view correspondability of features learned with (or without) these objectives, with a simple K-nearest-neighbors retrieval task. In Table 3, we show that (1) the 2D contrastive loss delivers more correspondable features than an RGB regression loss, and (2) using the 2D loss and 3D loss together yields the best results (even improving the 2D precision scores).

---

### Official Review · AnonReviewer3 · 2019-10-26
**Official Blind Review #3**

**Rating:** 6

**Review:**

The paper proposes a view-prediction inverse graphics model that takes input RGBD streams and produces a 3D feature map of the scene, including regions that are unobserved due to occlusion.  This model is used to learn a 3D visual representation that can be applied for semi-supervised 3D object detection, and for unsupervised 3D moving object detection.  The approach is similar to recent work on Geometry-aware RNNs (Tung et al. 2019), but is applied to more realistic scenes (urban landscapes datasets generated using the CARLA simulator as opposed to simple tabletop arrangements of ShapeNet objects) and makes fewer assumptions about the camera pose (6 DoF camera parameterization as opposed to 2 DoF cameras placed around the objects).  Moreover, the proposed model is evaluated on the downstream 3D object detection tasks to demonstrate the utility of the learned 3D visual representation.

Experiments are performed to evaluate the proposed method against baselines on 3D object detection (both in the semi-supervised setting and unsupervised moving object detection), 3D motion estimation, as well as sim-to-real transfer results (training in CARLA and testing on the KITTI dataset).  The results show that the proposed method: 1) has higher 3D object detection mAP than a view regression-based baseline from prior work (Tung 2019) for settings with little available 3D boundign box supervision; 2) has better 3D detection transfer from CARLA to KITTI (evaluated by mAP) compared to view regression baseline; 3) outperforms the view regression baseline and a 3D motion flow-based baseline on 3D moving object detection (measured through precision-recall curves and mAP); 4) has lower 3D flow error for moving objects compared to zero-motion and view regressive baselines.

I am positive with respect to accepting this work, but find that there are a few unclear points in the evaluation that should be clarified to strengthen the empirical results.  The data is generated from 50 frame sequences at 30fps (i.e. ~1.6 seconds of simulation) where each frame has 6 randomly sampled camera viewpoints in a 20m hemisphere in front of the car.  It would seem to me that the distances between the viewpoints in different frames from these sequences are likely to be quite small, so most of the viewpoint variance would come from random sampling within the hemisphere and randomly selecting one of the views as the target/unseen view. From this description, it is not clear how much variation of unobserved vs observed surfaces exists in the training and test data. It would have been informative to provide some statistics about observed vs occluded object surface area to elucidate the dataset construction.  This aspect of the dataset likely impacts the performance of the method significantly and should thus be addressed a bit more clearly.

**Experience Assessment:**

I have read many papers in this area.

**Review Assessment: Checking Correctness Of Derivations And Theory:**

I assessed the sensibility of the derivations and theory.

**Review Assessment: Checking Correctness Of Experiments:**

I assessed the sensibility of the experiments.

**Review Assessment: Thoroughness In Paper Reading:**

I read the paper at least twice and used my best judgement in assessing the paper.

---

> ### Author Response · Authors · 2019-11-12
> **Response to review**
>
> Dear reviewer,
>
> Thank you for your positive review. We have an answer for your concern:
>
>
> -- It would have been informative to provide some statistics about observed vs occluded object surface area to elucidate the dataset construction.
>
> You are right that this is an important and informative factor, since in a 2D or unrealistic 3D world, very little would be occluded. Based on your suggestion, we have calculated some statistics on this. For reference, we reiterate from the paper that the model encodes a volume of world space, which has dimensions 32 meters x 32 meters x 8 meters.
>
> A single camera view reveals information on 0.23 (±0.03) of all voxels in the space, leaving 0.77 totally occluded/unobserved. This measure includes both “scene surface” voxels, and voxels lying on rays that travel from the camera center to the scene surface (i.e., “known free space”). The revealed surface itself occupies only 0.01 (±0.002) of the volume. Adding a random second view, the total volume revealed is 0.28 (±0.05); the surface revealed is 0.02 (±0.003). With all 6 views, 0.42 (±0.04) of the volume is revealed; 0.03 (±0.004) is revealed surface.
>
> We have added these statistics to the appendix (Sec. C.1), to help support our claim that the vast majority of the scene must be “imagined” by the model during inference (as stated in Sec. 3.1).

---

### Official Review · AnonReviewer4 · 2019-11-01
**Official Blind Review #4**

**Rating:** 6

**Review:**

This paper deals with turning a 2.5D video representation into a 3D representation of an environment or a scene. The authors introduce self-supervised methods to pretrain the 2d-3d projection with a contrastive loss, which is the neural backbone for multiple other tasks. They then assess their approach on numerous tasks such as 3D-object detection, 3D-moving object detection, and 3D motion estimation. The authors also evaluate the transferability of the features in a challenging sim2real setting.

It is dense paper with multiple modules (2d-3d, ergomotion, memory, etc.) and concept. Still, the authors make it accessible by concise paragraphs, highlighting key equations (The enum + eq 1 and 2 are quite useful), and well-designed sketch (Figure1). I had some difficulties digging into the visual head component for each task as I was not familiar with this topic. However, the authors always explain their choices in a few lines and refer to the related papers for technical details in a meaningful way.

I am pretty convinced with the experiments, especially Sim2Real, in Tab1, where the baselines are clears and make sense.

I appreciated the limitation section, which is transparent and honest, and clearly states the strength and weaknesses (such as image downscaling) of the approach. Besides, the code and the data should be released, which is always a positive point.

Remarks:
 - Latent map update: running average is a simple and efficient mechanism, it also makes sense as you are dealing with big 3D tensors. Yet, have you tried other update mechanisms?
 - A natural follow-up to this paper is Contrastive Predictive Losses (which had several successes in pure vision setting[1]). Did you already try this approach?
 - In visual CPC papers [1] (or since the early days of visual representation learning!), data transformation has been applied to improve model performance. Would it make sense to apply it to I_{n+1}, D_{n+1} ?
 - Although the authors assess their approach with RGB-D, the models were still trained on 2.5D video. It would have been useful also to assess a pretraining on pure RGB-D data

However, I have two (somehow related) concerns. First of all, the machine learning novelties are rather small, contrastive losses are now widespread, and the models are closed to Tung et al. as mentioned by the authors.  However, I believe the paper to be a substantial contribution in vision, as they show the feasibility of their approach on large scale scenarios and over a highly diverse set of tasks. Again, the authors also release the code, making the paper a valuable baseline for the following work. On my side, I am impressed by the sim2real env.
My second concern is the following, it is a high quality vision paper, and I am curious why the authors chose ICLR over CVPR. Besides, the tasks are vision-oriented, and 2.5D vision is not common in the ML community. Having said that, the proposed approach is pretty generic, can be applied to RGB-D (more common in ML), and require few expert knowledge in vision (only the Egomotion module).

As a result, I would advocate for clear accept if we assess vision-based contribution for ICLR; otherwise, I would only recommend weak accept the paper is solely based on ML contributions (the paper is still sound, well-written, with numerous experiments and with a semi-generic architecture)

**Experience Assessment:**

I have read many papers in this area.

**Review Assessment: Checking Correctness Of Derivations And Theory:**

N/A

**Review Assessment: Checking Correctness Of Experiments:**

I assessed the sensibility of the experiments.

**Review Assessment: Thoroughness In Paper Reading:**

I read the paper at least twice and used my best judgement in assessing the paper.

---

> ### Author Response · Authors · 2019-11-12
> **Response to review**
>
> Dear reviewer,
>
> Thank you for your encouraging comments. Here are answers to your questions:
>
>
> -- Latent map update: running average is a simple and efficient mechanism, it also makes sense as you are dealing with big 3D tensors. Yet, have you tried other update mechanisms?]
>
> Yes, we tried a convolutional 3D GRU memory, but it did not give superior performance. This agrees with the findings of Tung et al., who mention that "averaging [...] works equally well to using the GRU update equations, while being much faster" (page 4 of that paper).
>
>
> -- A natural follow-up to this paper is Contrastive Predictive Losses (which had several successes in pure vision setting[1]). Did you already try this approach?
>
> Your reference is missing, but perhaps you are referring to "Representation Learning with Contrastive Predictive Coding" by Oord et al.? We agree, that is a good area for follow-up work. That paper uses a probabilistic contrastive loss, while we have a non-probabilistic contrastive loss. We elected to use the margin-based non-probabilistic version since it is simpler and better-explored in the literature (as you note also).
>
>
> -- In visual CPC papers [1] (or since the early days of visual representation learning!), data transformation has been applied to improve model performance. Would it make sense to apply it to I_{n+1}, D_{n+1} ?
>
> Yes, this makes sense, and we have actually tried it: we passed the RGB inputs through a 3-block ResNet to produce a 32-channel featuremap the same resolution as the RGB, and unprojected this to make a 32-channel 3D feature volume U. This added significant computational time and expense, without noticeably improving performance. We believe this is because the 3D encoder-decoder already has sufficient capacity to convert its raw inputs into informative features.
>
>
> -- Although the authors assess their approach with RGB-D, the models were still trained on 2.5D video. It would have been useful also to assess a pretraining on pure RGB-D data.
>
> We use the terms “2.5D video” and “sequences of RGB-D images” interchangeably in the paper. Our model takes K images as input and predicts view K+1. The number K can be one, in which case we just need two images viewing the same scene for training. In case you are wondering about "pure RGB data": with pure 2D RGB inputs, we find that performance deteriorates substantially. This suggests that the (sparse) 2.5D information input pointcloud is being used to good effect. (Please, let us know if we did not correctly interpret your comment).
>
>
> -- it is a high quality vision paper, and I am curious why the authors chose ICLR over CVPR. [...] I would advocate for clear accept if we assess vision-based contribution for ICLR; otherwise, I would only recommend weak accept the paper is solely based on ML contributions.
>
> Our paper demonstrates that view prediction can lead to learning semantically relevant visual features. We believe self-supervised visual learning is a very relevant topic for ICLR. After all, neuroscientists approximate that 80% of our sensory input comes from our eyes -- vision is essential for human-like AI and representation learning.
>
> We are partly inspired by the highly creative and impactful computer vision works that have been published at ICLR in the past (e.g., DeepLab from ICLR 2015, Learning Visual Predictive Models of Physics for Playing Billiards from ICLR 2016, Reinforcement Learning with Unsupervised Auxiliary Tasks in ICLR 2017, Spherical CNNs from ICLR 2018), and this year’s Call for Papers mentioning (among other things) feature learning, metric learning, large-scale learning, and applications in vision.
>
> We encourage you to weigh our paper’s contributions to computer vision in your final rating.

---

> > ### Comment · AnonReviewer4 · 2019-11-15
> > **Response**
> >
> > Thank you very much for your detailed response.
> >
> > I encourage the authors to add the remarks about "Latent map update", "data transformation" and other failed trials in the paper. They gave a lot of insight while reproducing/enhancing the results, and if they confirm other paper observation, it is still useful!
> >
> > After reading the rebuttal, I expect to increase my score in the upcoming discussions.

---

### Official Review · AnonReviewer2 · 2019-11-04
**Official Blind Review #2**

**Rating:** 3

**Review:**

This paper show that view prediction learning to help 3D detection. They explored the link of view predictive learning and the emergence of 3D perception in computational models of perception, on mobile agents in static and dynamic scenes. The whole paper is very well written, and organized.  In general, the whole model is quite straightforward, and quite heavy, while most components come from existing papers. It’s really a good engineering work in term of integrating them together. I have several comments.

how’s the proposed model different from [1], and [2]? Only Tung et al. (2019) is employed as baselines; and it’s really hard to tell whether the proposed model is SOTA.

(2) The model is still built upon Tung et al. (2019)  with several novel components, including handling more general camera motion beyond the a 2-degree-of-freedom sphere-locked camera.
I would like to check how significant this point? Particularly, how’s the performance of the model variant without using this component (just 2-degree-of-freedom sphere-locked camera) ?

(3) In  Fig.5, it is very interesting that, the results of  Estim.-stabilized 3D flow are even better than those of GT-stabilized 3D flow, when recall>0.5. Any insight here?

(4) The setting of SEMI-SUPERVISED LEARNING OF 3D OBJECT DETECTION is quite unclear and sloppy. What dataset are used as the labeled, and unlabelled images? how many labeled images?
Also not quite unclear about the settings in UNSUPERVISED 3D MOVING OBJECT DETECTION.

(5) Any chance to give some evaluation about the significance of components introduced in the model? This may give us more insights about the model. I notice that some experimental results may reflect some perspective of this point, while it may be better to explicitly discuss it.


[1]  Implicit 3D Orientation Learning for 6D Object Detection from RGB Images. ECCV 2018
[2]  PerspectiveNet: 3D Object Detection from a Single RGB Image via Perspective Points. NeurPIS 2019

**Experience Assessment:**

I have read many papers in this area.

**Review Assessment: Checking Correctness Of Derivations And Theory:**

I assessed the sensibility of the derivations and theory.

**Review Assessment: Checking Correctness Of Experiments:**

I assessed the sensibility of the experiments.

**Review Assessment: Thoroughness In Paper Reading:**

I read the paper thoroughly.

---

> ### Author Response · Authors · 2019-11-12
> **Response to review (part 1/2)**
>
> Dear Reviewer,
>
> Thank you for the criticisms. During this discussion period we hope to resolve each of your concerns and earn a higher rating.
>
> -- How is the proposed model different from [1], and [2]?
>
> Thank you for the references. In [1], the idea is to learn a denoising autoencoder for synthetic images of a particular object category, and then use the learned latent space to do nearest-neighbor look-up at test time, to transfer pose labels from the synthetic data to real images. To get object crops at test time (to get inputs for the autoencoder), they train a 2D detector supervised. Similar to our paper, a goal of that work is to reduce the number of real 3D labels required at test time. However, that model requires a ground-truth 3D mesh of each object of interest (to train the autoencoder), and also 2D boxes in the test domain (to train the detector). Our model, in contrast, only requires RGBD images and camera poses — without knowing where the objects are, or what they are. Nonetheless, [1] makes good strides on the sim-to-real problem and we can cite it as related work if you feel it is sufficiently close.
>
> In [2], the authors introduce a new 2D RCNN-like architecture for 3D object detection, where 2D detections are upgraded to 3D, making use of perspective points as an intermediate representation (and a local Manhattan World assumption) to simplify the 2D-to-3D problem. This is quite different from our work, in two major ways: (1) no part of it is self-supervised, and (2) it is image-centric, in the sense that information from a video sequence cannot be effectively integrated over time. In our paper we argue for self-supervision via multi-view metric learning, and our model makes use of video data: in static scenes we can integrate observations over time, and in dynamic scenes we can find 3D correspondences.
>
>
> -- Only Tung et al. (2019) is employed as a baseline; it is hard to tell whether the proposed model is SOTA.
>
> Thank you for mentioning this; we have clarified this in the paper. We focus on Tung et al. as a baseline because it has been empirically validated as the state-of-the-art. For example, it dramatically outperformed the highly influential DeepMind paper from last year (Eslami et al., Science, 2018). However, it is not our only baseline: in the appendix (Sec. C.3) we also evaluate the DeepMind model (GQN) and a carefully-tuned VAE variant of Tung et al., to measure whether those models produce representations that can be used for multi-view correspondence. We evaluate this with a simple K-nearest-neighbors task, both in 2D (with patches) and in 3D (with feature cubes). Our results show that only our contrastive objective delivers correspondability. This makes sense, because the contrastive objective is essentially a correspondence objective. We believe these correspondable features are critical for building a model that can produce flow, egomotion, and object detection, as we have done.
>
>
> -- How significant is it to use a 6-DoF camera instead of a sphere-locked 2-DoF one?
>
> This is an important question, and a critical part of the paper. To be able to test our model in CARLA and KITTI data, where the sensors are mounted on a moving vehicle, it is absolutely necessary to have a camera that can handle 3D translation. Tung et al. cannot handle this type of motion -- in fact, it can only be tested in settings where the data itself is sphere-locked. To evaluate our model against Tung et al., it was necessary to first upgrade their camera to 6-DoF.
>
>
> -- In Fig.5, estim.-stabilized 3D flow is better than GT-stabilized 3D flow, when recall>0.5. Any insight here?
>
> This is a very careful observation. Since the imperfect stabilization leaves some residual motion everywhere, it may be that the object motion is larger in this setting, making it easier for the 3D FlowNet to produce non-zero estimates in those regions, thereby leading to a few extra detections. However, this does not appear to be a very strong effect, and it may easily vanish with different flow training.
>
>
> (We continue in the next comment, due to the OpenReview character count limit.)

---

> > ### Author Response · Authors · 2019-11-12
> > **Response to review (part 2/2)**
> >
> > -- The settings of semi-supervised and unsupervised learning of object detection is quite unclear. What dataset are used as the labeled, and unlabelled images?
> >
> > Thank you for pointing this out; we can clarify:
> > We have 4 sets of data, which are train/val splits of CARLA and KITTI:
> > - CARLA train set: 124256 random samples from “City1”
> > - CARLA val set: 48268 random samples from “City2”
> > - KITTI train set (official split): 3712 examples
> > - KITTI val set (official split): 3769 examples
> >
> > We use this data in the following way:
> > - In the semi-supervised setting, we use the CARLA train set for view prediction training (without using box labels), and a randomly-sampled subset of the CARLA train set for box supervision. We varied the size of the box supervision subset, across the following range (to produce Figure 1): 100, 200, 500, 1000, 10000, 80000.
> > - In the sim-to-real setting, we use the CARLA train set for view prediction training (without using box labels), and the KITTI train set for box supervision. We evaluate on the KITTI val set.
> > - In the unsupervised setting, we train on the CARLA train set (without using box labels), and evaluate on the CARLA val set.
> >
> > We have added this information to the main text. (Previously some details were in the supplemental.)
> >
> >
> > -- Any chance to give some evaluation about the significance of components introduced in the model? This may give us more insights about the model. I notice that some experimental results may reflect some perspective of this point, while it may be better to explicitly discuss it.
> >
> > Yes, we can clarify this as well. Here are the introduced components, along with what they make possible:
> > - General camera motion, which allows us to test on challenging domains like CARLA and KITTI.
> > - Our metric learning loss, which gives us a superior feature representation, as validated in downstream tasks such as flow and object detection.
> > - Our 3D flow net, which allows a simple unsupervised 3D object discovery method. To the best of our knowledge, this is the first work that can discover objects in 3D from a single camera viewpoint, without any human annotations of object boxes or masks.
> >
> >
> > Thank you again for all of your constructive criticism. We have updated the pdf according to your suggestions. We hope that you will soon increase your rating and/or suggest additional ways we can improve.

---

### Author Response · Authors · 2019-11-12
**General response**

We thank each reviewer for their detailed feedback.

We have updated our manuscript with the requested additions and clarifications. This includes:
- In Sec. 3.1, clarification on why 6-DoF cameras are critical.
- In Sec. 3.1 and Sec. 4, a short discussion of the 2D and 3D correspondence experiments (present in Sec. C.3 in the appendix), in which we evaluate ablations of our model and additional related work.
- In Sec. 4, a detailed description of the dataset and training procedure for each problem setting.
- In Sec. B (appendix), a clarification of the 3D-to-2D projection module.
- In Sec. C.1 (appendix), detailed occlusion/visibility statistics from the data.
- In Sec. C.2 (appendix), a discussion and visualization of the baselines' RGB view prediction outputs on CARLA.

To make these edits easy to find, we have formatted them with red text. These changes have increased the length of the paper to 8.5 pages, but we should be able to edit back down to 8 if necessary.

Please let us know if there are additional questions or concerns. We will closely monitor this forum until the discussion period ends.

---

### Decision · Program_Chairs · 2019-12-19

**Decision:**

Accept (Poster)

**Comment:**

The authors propose to learn space-aware 3D feature abstractions of the world given 2.5D input, by minimizing 3D and 2D view contrastive prediction objectives. The work builds upon Tung et al. (2019) but extends it by removing some of the limitations, making it thus more general. To do so, they learn an inverse graphics network which takes as input 2.5D video and maps to a 3D feature maps of the scene. The authors present experiments on both real and simulation datasets  and their proposed approach is tested on feature learning, 3D moving object detection, and 3D motion estimation with good performance. All reviewers agree that this is an important problem in computer vision and the papers provides a working solution. The authors have done a good job with comparisons and make a clear case about their superiority of their model (large datasets, multiple tasks). Moreover, the rebuttal period has been quite productive, with the authors incorporating reviewers' comments in the manuscript, resulting thus in a stronger submission. Based in reviewer's comment and my own assessment, I think this paper should get accepted, as the experiments are solid with good results that the CV audience of ICLR would find relevant.